# Combined Reforming of Clean Biogas over Nanosized Ni–Rh Bimetallic Clusters

**Nicola Schiaroli** [1,*] **, Carlo Lucarelli** [2,*] **, Maria Carmela Iapalucci** [1] **, Giuseppe Fornasari** [1] **, Antonio Crimaldi** [1] **and Angelo Vaccari** [1,*]

1   Dipartimento di Chimica Industriale "Toso Montanari", Alma Mater Studiorum, Università di Bologna, Viale del Risorgimento 4, 40136 Bologna, Italy; maria.iapalucci@unibo.it (M.C.I.); giuseppe.fornasari@unibo.it (G.F.); antonio.crimaldi@studio.unibo.it (A.C.)
2   Dipartimento di Scienza e Alta Tecnologia, Università dell'Insubria, Via Valleggio 9, 22100 Como, Italy
*   Correspondence: nicola.schiaroli@unibo.it (N.S.); carlo.lucarelli@uninsubria.it (C.L.); angelo.vaccari@unibo.it (A.V.)

**Abstract:** The combined steam/dry reforming of clean biogas ($CH_4/CO_2$ = 50/50 *v/v*) represents an innovative way to produce synthesis gas (CO + $H_2$) using renewable feeds, avoiding to deplete the fossil resources and increase $CO_2$ pollution. The reaction was carried out to optimize the reaction conditions for the production of a syngas with a $H_2/CO$ ratio suitable for the production of methanol or fuels without any further upgrading. Ni-Rh/Mg/Al/O catalysts obtained from hydrotalcite-type precursors showed high performances in terms of clean biogas conversion due to the formation of very active and resistant Ni-Rh bimetallic nanoparticles. Through the utilization of a $\{Ni_{10}Rh(CO)_{19}\}\{(CH_3CH_2)_4N\}_3$ cluster as a precursor of the active particles, it was possible to promote the Ni-Rh interaction and thus obtain low metal loading catalysts composed by highly dispersed bimetallic nanoparticles supported on the MgO, $MgAl_2O_4$ matrix. The optimization of the catalytic formulation improved the size and the distribution of the active sites, leading to a better catalyst activity and stability, with low carbon deposition with time-on-stream.

**Keywords:** combined steam/dry reforming; biogas; Ni–Rh catalyst; cluster; synthesis gas

## 1. Introduction

With the aim of achieving the reduction of greenhouse gas emissions, the production of biogas (BG) has increased in the last decades, leading to a more efficient management of wastes [1,2]. Different types of biomass residues can be used to generate biogas such as landfills [3,4], sewage sludge [5–7] or biowaste digesters [7–9] with a variable chemical composition [10]. Nowadays BG production and upgrading plants are spread all around the world; in Europe they are located mainly in Germany, the United Kingdom, Italy, the Czech Republic and France [11–13]. When purified from main pollutants, the clean biogas (CB) obtained is a mixture of $CH_4$ and $CO_2$ [14] that can be used as such to produce electricity [15,16], heat [17,18] or after $CO_2$ removal directly introduced in energy grid [19–21]. Syngas represents a crucial feedstock that can be converted into methanol or liquid fuels (Fischer–Tropsch synthesis). It is produced on a large scale by the conversion of fossil fuels, including methane reforming, coal gasification and heavy oil partial oxidation. The depletion of fossil fuels, together with the environmental drawbacks associated with the utilization of these sources, has in the last decade promoted the conversion of CB to synthesis gas (or syngas, CO + $H_2$) [22,23]. Even if the $CO_2$ removal and the utilization of biomethane (BM) to produce syngas is feasible, the full valorization of both $CH_4$ and $CO_2$ is more desirable considering their negative environmental impact [19–21]. Hence, the necessity of finding a way to convert them into a cleaner form of energy is apparent [24].

In this sense, CB reforming—converting $CO_2$ along with $CH_4$ into syngas—is a strategy that allows the production of more sustainable fuels and valuable chemicals through renewable sources. It is attractive to convert CB in the industrial steam reforming (SR) ($CH_4 + H_2O \leftrightharpoons CO + 3H_2$ $\Delta H^r_{298K} = +206$ kJ/mol) existing plants, but the steam rich condition inhibits the $CO_2$ conversion reducing the productivity of the process [25,26]. Dry reforming (DR) ($CH_4 + CO_2 \leftrightharpoons 2CO + 2H_2$ $\Delta H^r_{298K} = +247$ kJ/mol) represents the best route to valorize the CB, but the very quick catalyst deactivation due to the fast coke formation limits its commercialization [27–32]. Combining SR and DR and modulating the amount of steam currently represents the best compromise for converting $CO_2$ and limiting the catalyst deactivation. The combined steam/dry reforming process is also possible via use of the established industrial technologies, avoiding huge investments. Different classes of catalysts have been investigated for this reaction, but Ni-based materials are still the most used in industrial applications, due to the good compromise between the activity and price [33–38]. Noble and transition metals have been investigated for reforming reactions, but the high activity and the resistance to the deactivation of noble metals does not justify their application on a large scale, due to the production costs [39–42]. Adding little amounts of noble metals as promoters to the Ni-based catalysts formulation represents the best route to improve the catalytic performances, limiting the increases of investments [43–51]. In the present work, the valorization of CB to produce a syngas with a $H_2$/CO ratio of approximately 2 that is directly convertible in methanol or fuels was deeply investigated. Starting from previous studies [52,53], in which the role of Rh as the dopant for the Ni/Mg/Al catalysts from hydrotalcite-type (HT) precursors was evidenced, a series of catalysts prepared by impregnation of a Ni–Rh cluster on a Mg/Al/O mixed oxide (Mg/Al = 4 as atomic ratio, obtained by calcination of a HT precursor) were studied in order to improve the interaction between the two metals promoting the formation of alloys that enhance the activity, strongly reducing the deactivation. Based on the results obtained, the reduction of the active phase amount was investigated with the aim of a cost reduction, proposing a catalytic formulation of possible interest for industrial scale-up.

## 2. Results and Discussion

### 2.1. Characterization of Fresh Catalysts

The catalyst support was obtained by coprecipitation of a Mg/Al HT precursor [52] that resulted in a mixture of MgO and $MgAl_2O_4$ after calcination at 900 °C, as shown by the X-ray diffraction (XRD) pattern (Figure 1, left).

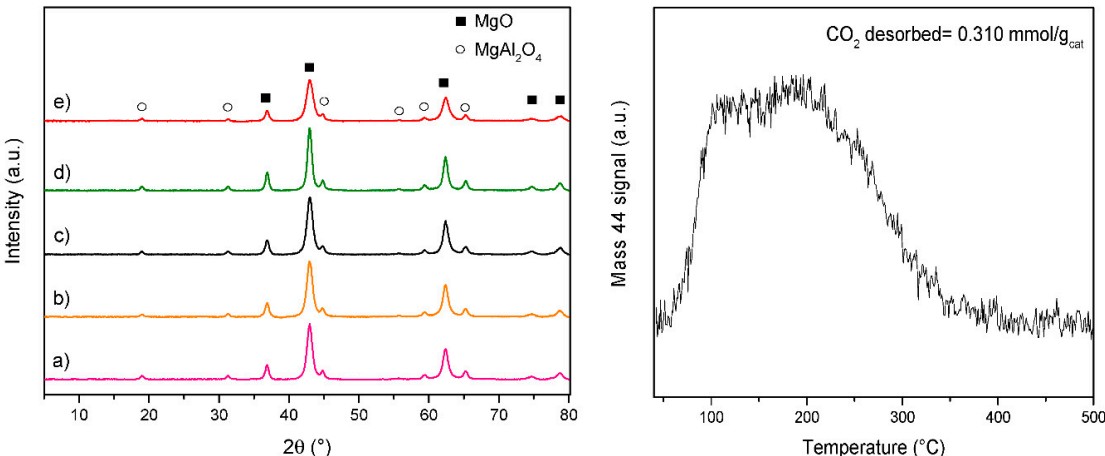

**Figure 1.** On the left, XRD patterns of the catalysts after calcination at 900 °C: (a) Mg/Al/O catalyst support, (b) 2NiRh C, (c) 3NiRh C, (d) 3NiRh IWI, (e) 3NiRh CP. On the right, $CO_2$-TPD of the catalyst support. C: cluster; WI: wet impregnation; IWI: incipient wetness impregnation; CP: coprecipitation of a hydrotalcite-type precursor.

The $CO_2$-TPD (temperature-programmed desorption) analysis (Figure 1 right) showed a broad $CO_2$ desorption peak between 70 and 350 °C, that can be divided in three distinct regions as a function of the strength of the superficial basic sites [54,55]. The region between 100–250 °C revealed the spread presence of weak and medium basic sites due to the formation of bicarbonates and bidentate carbonates species desorbed from $Mg^{2+}$-$O^{2-}$ pairs, respectively, while the low fraction of $CO_2$ desorbed at a T > 250 °C was likely related to the presence of strong basic sites ($O^{2-}$) that led to the formation of stabilized monodentate carbonates.

After the impregnation of the catalyst support by $\{NEt_4\}_3\{Ni_{10}Rh(CO)_{19}\}$ and thermal decomposition at 160 °C, as displayed by EDX (Energy-dispersive X-ray spectroscopy) analyses on different spots (Figure 2), the solid was totally covered by small metal particles dispersed over the support surface. The compositional profiles of the solid confirmed the co-presence of Ni and Rh revealing an average Ni/Rh atomic ratio between 5 and 11.

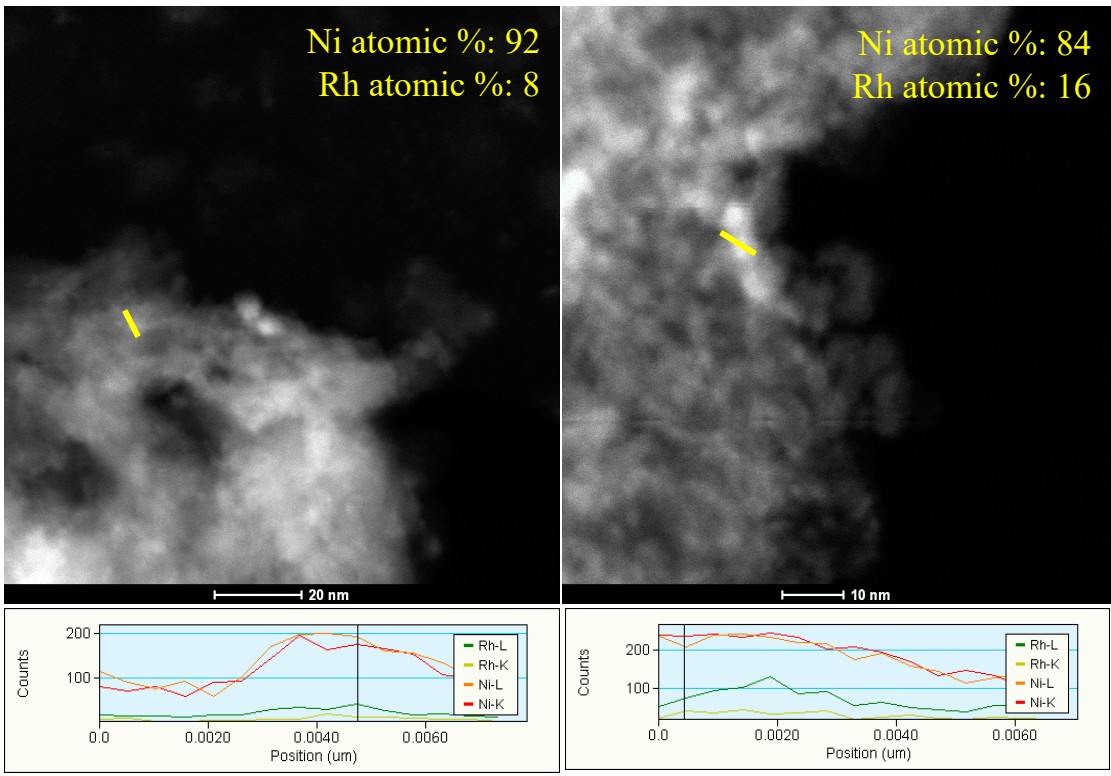

**Figure 2.** EDX profile analyses on two spots of the 5NiRh C catalyst.

After calcination, the 2NiRh cluster (C) and 3NiRh C catalysts exhibited comparable specific surface areas of ~80 $m^2$/g (Table 1), while the increase of the metal loading led to a dramatic decrease to 48 $m^2$/g in the case of 5NiRh C sample. A slight decrease of specific surface area was also observed for the catalyst obtained by incipient wetness impregnation (IWI) (60 $m^2$/g).

**Table 1.** Specific surface area (SS) of the catalyst before and after reaction.

| $SS_{BET}$ | 2NiRh C | 3NiRh C | 5NiRh C | 3NiRh IWI | 3NiRh CP |
|---|---|---|---|---|---|
| Before reaction ($m^2$/g) | 73 | 80 | 48 | 60 | 99 |
| After reaction ($m^2$/g) | 43 | 39 | 39 | 26 | 46 |

Independently from the method used for the incorporation of Ni and Rh, after calcination at 900 °C, all the Ni–Rh/Mg/Al catalysts were composed by almost the same crystalline phases. The XRD patterns of the samples (Figure 1 left) showed the presence of an intense series of peaks

related to MgO, according to the high $Mg^{2+}/Al^{3+}$ ratio of the catalyst support (4.0 as atomic ratio) together with the formation of a spinel phase composed mainly by Mg and Al ($MgAl_2O_4$). The peaks related to the presence of Ni and Rh oxides were not observed. It is noteworthy that, as previously reported [52,56], the formation of Mg/Ni/O and $Ni(Mg)Al(Rh)_2O_4$ phases very likely occurred at such high temperature, consequently leading to the development of strong metal–support interactions (SMSI) under reducing conditions.

The $H_2$-TPR (temperature-programmed reduction) experiments of the calcined catalysts performed over a broad range of temperatures are reported in Figure 3. The peak located in the lower temperature region was assigned to the reduction of highly stabilized $Rh^{3+}$ species, that occurred in the 350–400 °C range. Notably, this peak became almost negligible for the 5NiRh C sample (Figure 3), suggesting the presence of larger $Rh^0$ particles deposited from the metal cluster used, difficult to be oxidized and/or reduced. Conversely, the moderate Rh activation observed in the co-precipitated catalyst (3NiRh CP), was likely related to a partial presence of Rh species in the bulk phase. The peak detected around 900 °C was attributed to the reduction of $Ni^{2+}$ species stabilized by the MgO matrix or partially present as $NiAl_2O_4$ phase. Comparing the results obtained for catalysts with a nominal Ni load of 3.0 wt % (Figure 3) it is possible to observe that the reducibility in this region was easier for the 3NiRh C and 3NiRh CP catalysts, suggesting stronger interaction between Ni and Rh, that catalyzes the reduction of $Ni^{2+}$ species by $H_2$-spillover, decreasing its reduction temperature. The formation of metal nanoparticles was confirmed by the HRTEM images of the catalysts after the $H_2$-TPR experiments (Figure 4), showing the presence of small and highly dispersed metal particles (Figure 4a,b,d) embedded in or deposited on the MgO and $MgAl_2O_4$ matrix (Figure 4c). The average size of the formed nanoparticles was comparable among the catalysts at around 2–3 nm; notably the 3NiRh C catalyst showed a marked and spread presence of active species located in kink positions, and was thus reasonably more active in the reaction.

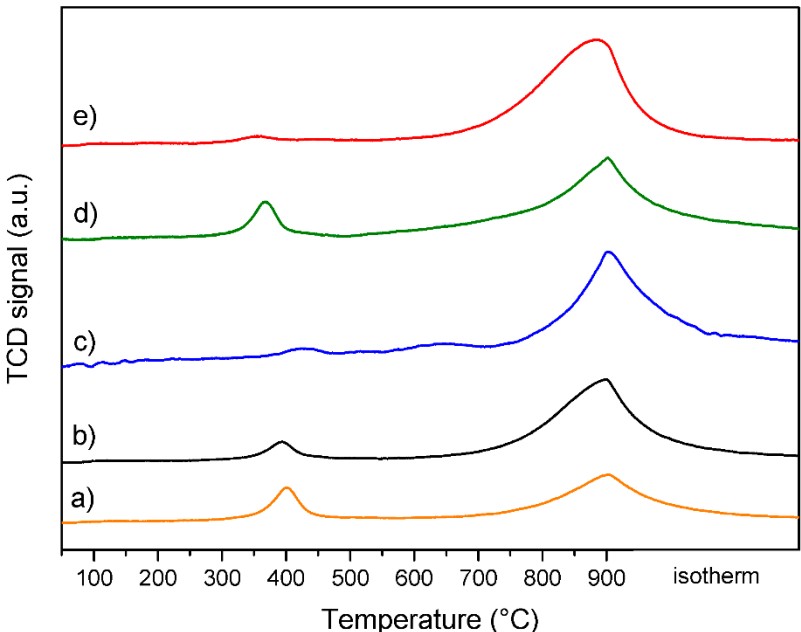

**Figure 3.** $H_2$-TPR profiles of the investigated catalysts: (a) 2NiRh C, (b) 3NiRh C, (c) 5NiRh C, (d) 3NiRh IWI, (e) 3NiRh CP.

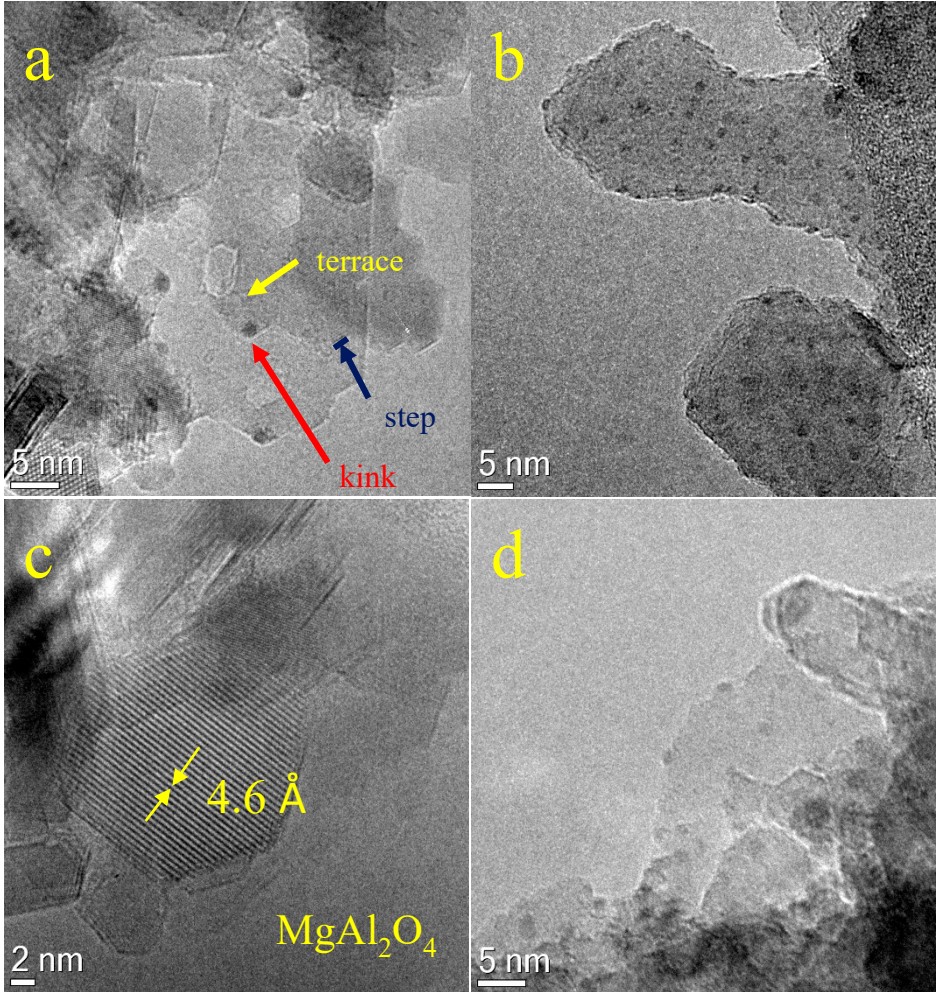

**Figure 4.** HRTEM images of the investigated catalysts after $H_2$-TPR analysis: (**a**) 3NiRh C, (**b**) 2NiRh C, (**c**,**d**) 3NiRh IWI.

## 2.2. Combined Steam and Dry Reforming of Clean Biogas

The performances of the catalysts were studied at decreasing temperature steps, varying the amount of steam in the inlet stream (Figure 5a,b). Hence, it was possible to evaluate the conditions in which CB could be better valorized. The biogas conversion values calculated at the thermodynamic equilibrium (red lines in Figure 5) showed that the $CH_4$ conversion was less affected by the amount of water fed. As expected, by decreasing the $S/CH_4$ ratio, the experimental $CO_2$ conversion was highly enhanced, while the $CH_4$ conversion lightly decreased. Almost every catalyst tested at 900 °C achieved $CH_4$ conversion values close to thermodynamic equilibrium, while the $CO_2$ conversions approached the equilibrium values only at 900 °C, $S/CH_4 = 0.5$. With a low metal loading (2NiRh C catalyst) it was possible to obtain high conversions of $CH_4$ at 900 °C even at low $S/CH_4$ ratio ($CH_4$ conversion = 86% at $S/CH_4 = 0.5$), where over the 65% of $CO_2$ was also converted. Decreasing the reaction temperature to 800 °C, 2NiRh C turned to be sensibly less active especially in terms of $CO_2$ conversion, with a poor result of 10% at $S/CH_4 = 2.0$. When the Ni–Rh cluster was impregnated to obtain a Ni load of 3.0 wt % (3NiRh C), the catalyst exhibited superior catalytic performances in all the reaction conditions investigated. Remarkably, 70% of the $CO_2$ fed was converted at 900 °C and almost 90% of the $CH_4$ was valorized even at 800 °C ($S/CH_4 = 2.0$).

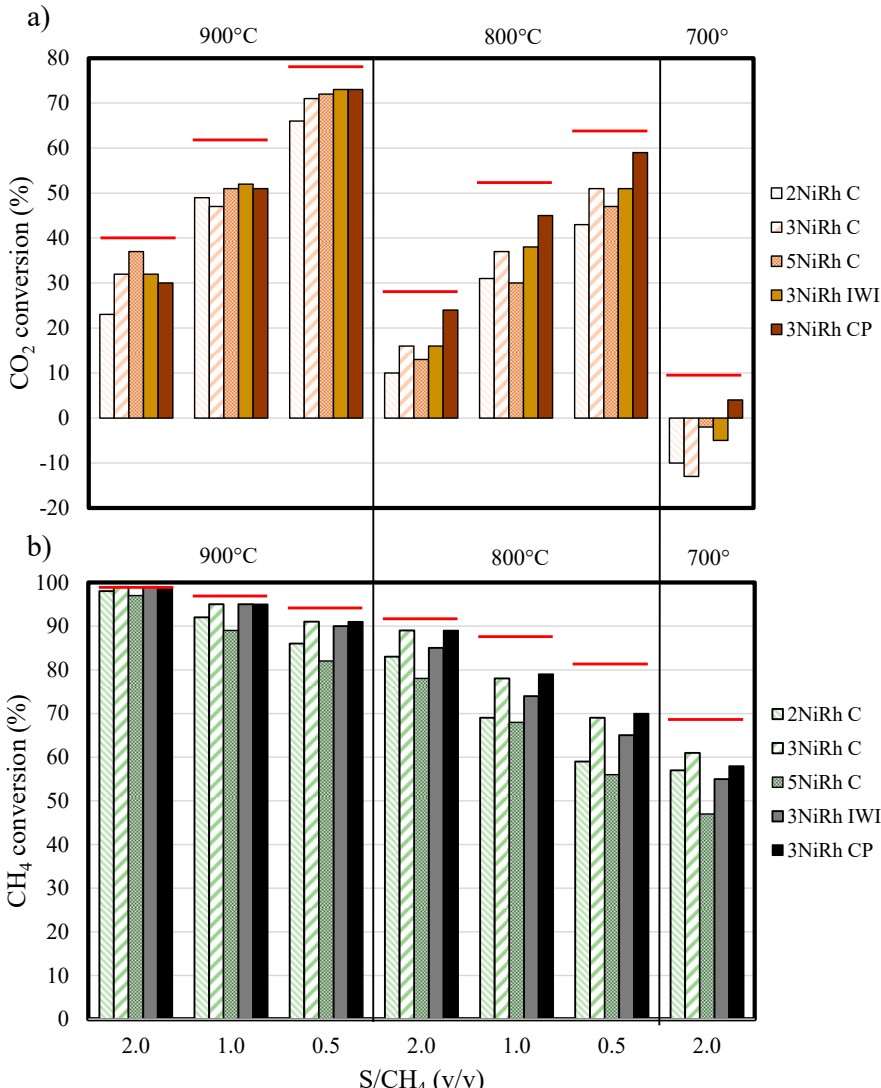

**Figure 5.** $CO_2$ (**a**) and $CH_4$ (**b**) conversion values as a function of the reaction temperature and $S/CH_4$ ratio values in the inlet stream. The red lines show the theoretical $CH_4$ or $CO_2$ conversion values calculated at the thermodynamic equilibrium.

Surprisingly, increasing the metal loading to 5.0 wt % of Ni (Rh = 0.9 wt %) the catalytic performances drastically decreased, reaching the lowest values of CB conversion among the tested systems, e.g., at 700 °C most of the $CH_4$ remained unreacted. The "volcano trend" observed with the increase of the amount of active phase may be explained considering the poor 5NiRh C catalyst activation in reducing conditions ($H_2$-TPR, Figure 3). Although the larger and stable $Rh^0$ particles obtained after calcination were active for the reaction, the synergic interactions between Ni–Rh were not strong enough to promote the activity of the Ni-species, hardly reducible and thus less active in the combined reforming.

To make a comparison between the promising activity of these novel catalysts with those attainable incorporating the active phase through the largely employed IWI and coprecipitation of a hydrotalcite-type precursor (CP) techniques, two catalysts with the same composition of the 3NiRh C sample were prepared and labelled as 3NiRh IWI and 3NiRh CP, respectively. Remarkably, 3NiRh C outperformed the 3NiRh IWI catalyst in terms of $CH_4$ conversion at 800 and 700 °C, with no significant differences in the amount of $CO_2$ converted. The possibility to valorize the CB using catalysts with low metal loadings was further confirmed by the 3NiRh CP sample. Starting from the results obtained in our previous work [52] in which the performances of different Ni-based catalysts obtained by CP

were assessed (Ni loading of 10 wt %), the amount of active phase was reduced and the catalytic results attained (Figure 5) showed a remarkable improvement of activity especially in terms of $CO_2$ conversion, while the amount of $CH_4$ processed was comparable to the one displayed by the 3NiRh C catalyst.

It is noteworthy that, for all the catalyst tested, the occurrence of the water gas shift (WGS) reaction was not negligible at 700 °C. The excess of steam in the inlet stream drastically reduced the $CO_2$ conversion, which turned towards negative values in most of the cases (with the exception of the 3NiRh CP catalyst, c.a. 5%). This behavior was confirmed by the high $H_2/CO$ values of the syngas produced (~3 as molar ratio at 700 °C). Repeating the test at 900 °C ($S/CH_4 = 2$) at the end of the cycle of experiments no loss of activity was detected, evidencing the high stability of the different investigated catalysts also in harsh reaction conditions.

The productivity of the process in the different operating conditions was calculated by considering the overall volumes of CB converted (as the sum of $CO_2$ and $CH_4$ valorized) expressed as $L_{BGconv.}$ $h^{-1}$ $g_{cat}^{-1}$ (Figure 6). The competition between the two main reforming reactions led to significant differences regarding the composition of the outlet syngas as well as the amount of CB processed. Depending on the amount of water fed, it was possible to modulate the $H_2/CO$ ratio in the outlet stream (Figure S3). Notably, when the $S/CH_4$ value was 0.5, high volumes of CB were converted, achieving the 82% of the total fed at 900 °C and producing a syngas with a $H_2/CO$ ratio of 1.2 (*v/v*). Conversely, in excess of steam, the amount of CB processed slightly decreased (65% at 900 °C and 55% at 800 °C), but the significant occurrence of the SR and WGS reactions enriched the $H_2$ content in the syngas, that exhibited $H_2/CO$ values in the range 2–2.3 (*v/v*). Such a wide variety of syngas compositions was attainable by modulating the reaction conditions, further evidencing the potentiality of the combined reforming to valorize a large volume of CB without using further steps in its upgrading process, thus producing a syngas exploitable in downstream applications such as Fischer–Tropsch or methanol synthesis. Finally, the results obtained in the different conditions were normalized on the amount of catalytic active phase loaded (Ni + Rh), expressing the results in terms of volume of CB converted per hour and gram of active phase ($L_{(CH4 + CO2)}$ $h^{-1}$ $g_{(Ni + Rh)}^{-1}$). The obtained values were compared to the biogas conversions previously reported in the literature, in Table 2.

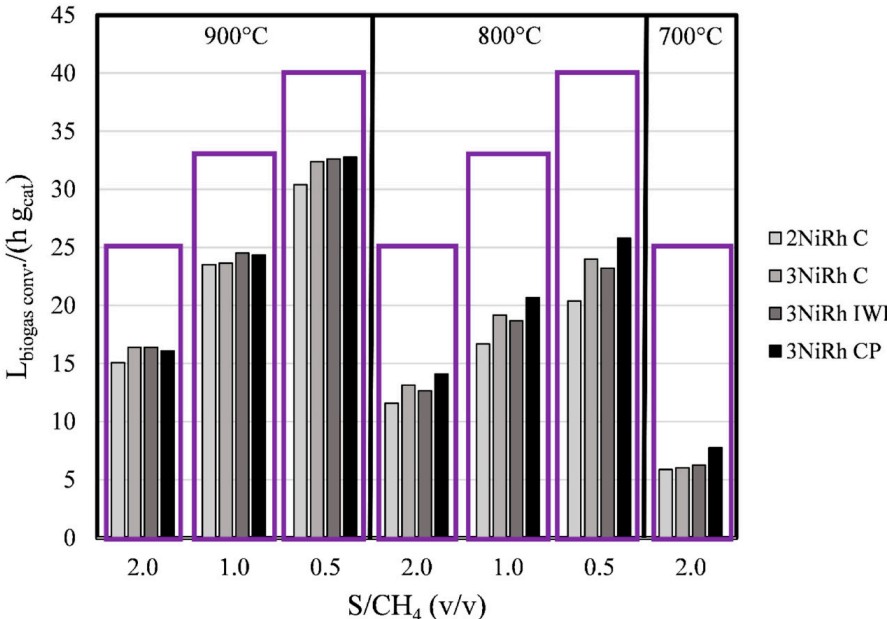

**Figure 6.** Conversion of clean biogas for the different temperatures and $S/CH_4$ values. The purple rectangles show the liters of clean biogas fed in each test condition.

**Table 2.** Comparison between the amount of biogas converted in this work (expressed as $L\ h^{-1}\ g_{active\ phase}^{-1}$) and the results obtained for different Ni-based catalysts reported in literature.

| Catalyst Active Phase/Promoter/Support | GHSV (mL $h^{-1}$ $g^{-1}_{cat}$) | P (Mpa) | T (°C) | S/CH$_4$ (v/v) | Biogas Converted (L $h^{-1}$ $g_{active\ phase}^{-1}$) | Ref. |
|---|---|---|---|---|---|---|
| 3NiRh C | 50,000 | 0.5 | 900 | 0.50 | 926 | this work |
|  |  |  | 800 | 0.50 | 686 |  |
|  |  |  | 700 | 2.00 | 172 |  |
| Ni (10 wt %)/La$_2$Zr$_2$O$_7$ | 20,000 | 0.1 | 700 | 1.00 | 68 | [57] |
| Ni (10 wt %)/La$_2$Zr$_2$O$_7$ | 60,000 | 0.1 | 700 | 1.00 | 94 | [57] |
| Ni (10 wt %)/Al$_2$O$_3$ | 12,000 | 0.1 | 700 | 1.20 | 17 | [38] |
| Ni (15 wt %)/MgO | 60,000 | 0.7 | 830 | 0.80 | 181 | [58] |
| Ni (4 wt %)/ Rh$_2$O$_3$ (0.04 wt %)/γ-Al$_2$O$_3$ | 6000 | 0.1 | 750 | 0.67 | 76 | [59] |
| Ni (5 wt %)/γ-Al$_2$O$_3$ | 69,000 | 0.1 | 800 | 0.80 | 731 | [60] |
| Ni (15 wt %)/ RuO$_2$ (0.5 wt %)/HDL | 120,000 | 0.1 | 750 | 0.14 | 231 | [61] |
| Ni (10 wt %)/ Mo-carbide (0.5 wt %)/ZrO$_2$ | 60,000 | 0.1 | 850 | 0.80 | 308 | [62] |

Considering the harsh reaction conditions used in this work, especially in terms of operating pressure and S/CH$_4$ values, it was possible to assert that the catalysts tested were promising for the combined reforming of biogas even at low temperature. It is noteworthy that, even at 700 °C, the value of biogas converted per gram of active phase was highly competitive to that observed for other Ni-based catalysts, reaching an outstanding value of 926 L $h^{-1}$ $g_{(Ni + Rh)}^{-1}$ at 900 °C when a S/CH$_4$ of 0.50 was used.

## 2.3. Characterization of Spent Catalysts

To shed light on the reasons behind the different catalytic performances and to correlate the properties of the catalysts with the method of active metal incorporation, the catalysts were deeply characterized after reaction by HRTEM, XRD, BET and Raman analyses. The harsh reaction conditions used caused a loss of specific surface area for all the tested catalysts (Table 1) that reached a value down to 26 m$^2$ g$^{-1}$ for the 3NiRh IWI sample. Although this decrease also caused an increase of crystallinity in the catalysts, the XRD patterns of the spent catalysts (Figure S4) did not evidence any peak related to the presence of Ni$^0$ or Rh$^0$ phases, testing the good dispersion and stability of the active species. From the HRTEM and HAADF (high-angle annular dark field) images (Figure 7) it was possible to determine the statistical distribution of the metal particles and the composition of them as a function of their dimension.

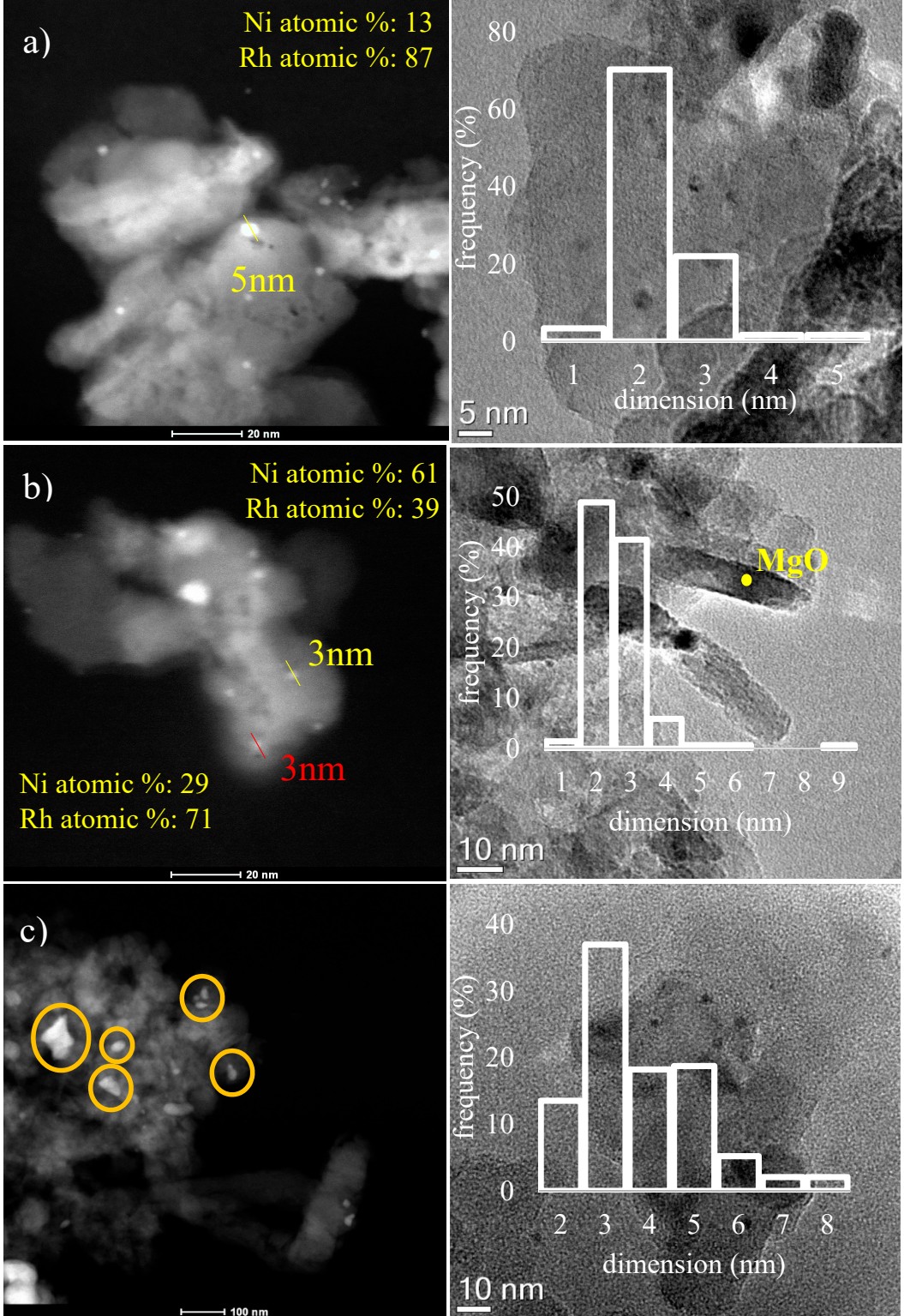

**Figure 7.** HAADF and HRTEM images with particle size distribution and Ni, Rh content for the evidenced areas of the catalysts after reaction. (**a**) 2NiRh C, (**b**) 3NiRh C, (**c**) 5NiRh C.

To some extent, all the tested catalysts showed two distinct particles domains that were characterized by a different distribution between small and highly active nanoparticles and the formation of larger agglomerates (d > 20 nm), whose presence significantly varied among the samples. The best active phase dispersion was achieved for the 2NiRh C catalyst (Figure 7a), that exhibited an

average diameter of the formed nanoparticles around 2.5 nm. Although the presence of some particles with a larger diameter of ~16 nm was detected, the active phase was well distributed on the surface of the catalyst and exhibited the co-presence of Ni and Rh, confirming the stability of the bimetallic alloy formed during reaction/reduction. The average particle diameter varied as a function of the metal loading of the catalysts, consequently affecting the statistical distribution of the active sites and consequently the catalytic activity. The 3NiRh C catalyst (Figure 7b) displayed a narrow distribution of the Ni–Rh nanoparticles centered around 2–3 nm, while when the metal loading increased, the particle distribution became wider, further revealing a significance presence of active particles with a diameter higher than 4 nm (~50% of the total). Furthermore, also due to the low surface area displayed after calcination (48 m$^2$/g), a widespread presence of large sintered metal aggregates (up to ~100 nm) composed by Ni and Rh was observed (Figure 7c). The formation of such large particles was likely the responsible for the loss of activity observed during the catalytic tests. Similarly, for the 3NiRh IWI catalyst (Figure 8), the formation of small Ni–Rh particles (with a distribution centered around 3 nm) occurred together with a conspicuous presence of larger particles with a diameter between 10 and 20 nm.

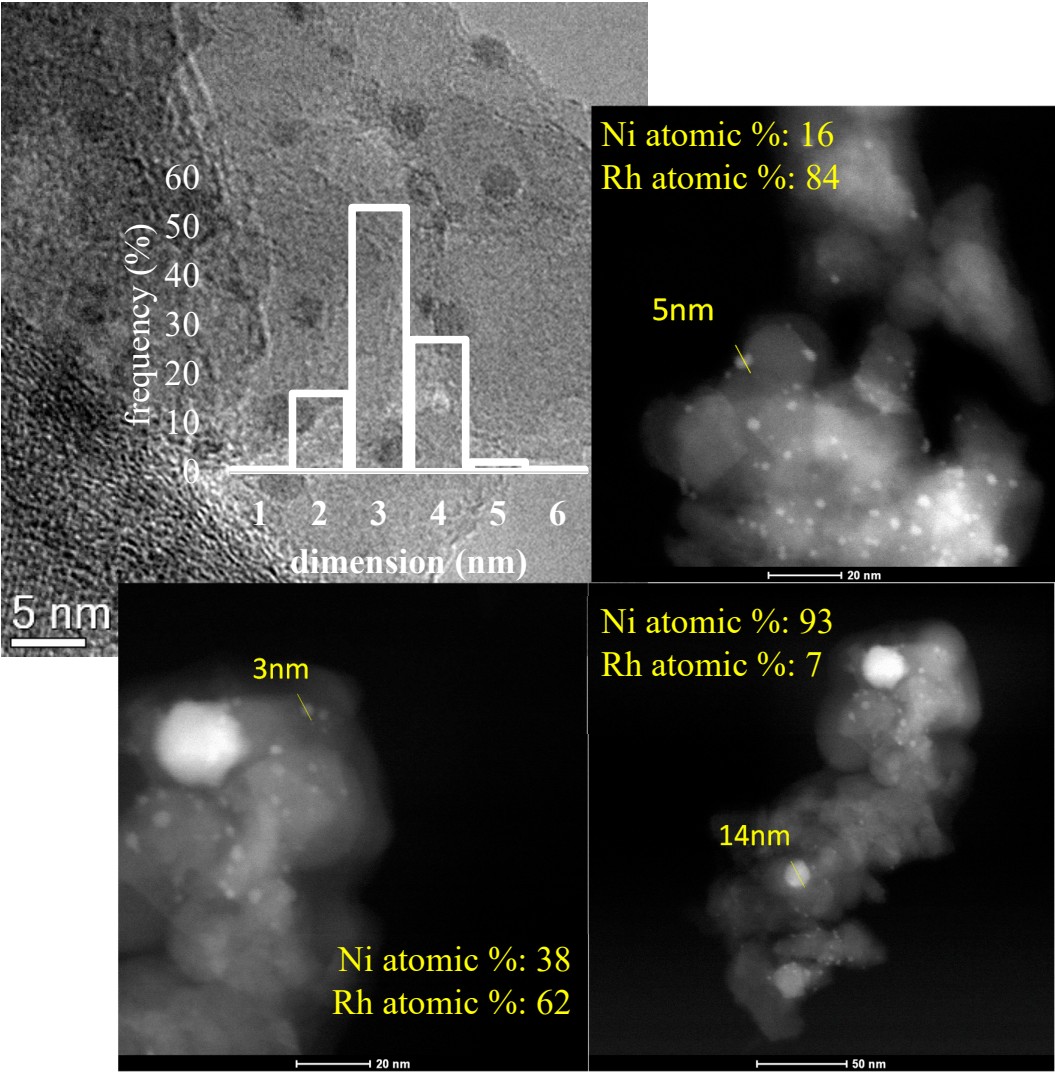

**Figure 8.** HRTEM and HAADF images, particle size distribution and EDX analyses for different particles of the 3NiRh IWI catalyst after reaction.

Comparing the results obtained at equal metal loading, it is possible to assert that the type of precursor used for the Ni–Rh incorporation process significantly affected the formation of the nanoparticles and their resistance to sintering phenomena during the catalytic tests. These properties were improved using the $Ni_{10}Rh(CO)_{19}$ cluster that assured a better interaction between Ni and Rh, promoting the formation of highly active sites.

Regarding the composition of the nanoparticles, the overall Ni and Rh content varied as a function of the particle dimension. For example, the active particles in the 3NiRh C catalyst were composed by variable amounts of the two metals, while the smaller nanoparticles showed a high content of Rh (30–70 as atomic%), the larger particles (>9 nm) formed by Ni sintering are Ni enriched and showed a negligible Rh content. The same trend was observed for the other catalysts, confirming that the presence of Rh promoted the stabilization of the $Ni^0$ particles by the formation of a bimetallic alloy. Further investigating the carbon formation during reaction, Raman spectra on different spots of the catalysts were recorded (Figure S5). For all tested systems, the analyses did not show a widespread presence of carbonaceous material, as it is locally lightly deposited (mainly as nanotubes) over the 3NiRh C catalyst surface only, showing the characteristic G and D bands attributable to the formation of graphite and defects, respectively. It is noteworthy that this particular result can be attributable to the remarkable activity of the catalyst, that can at same extent allow also the $CH_4$ decomposition, especially at 900 °C and at low $S/CH_4$ ratios.

## 3. Material and Methods

### 3.1. Preparation of the Catalysts

The chemicals used for the synthesis were rhodium (III) nitrate solution (~10 wt % (Rh in >5 wt % $HNO_3$), Sigma Aldrich, St. Louis, MO, USA), nickel (II) nitrate hexahydrate (Sigma Aldrich, 99%), magnesium nitrate hexahydrate (Sigma Aldrich, 99%) and aluminum nitrate nonahydrate (Aldrich Sigma, 98%). The coprecipitation of the hydrotalcite type precursors were carried out by preparing a solution of $Ni^{2+}$, $Mg^{2+}$, $Al^{3+}$, $Rh^{3+}$ nitrates $((Ni,Mg)^{2+}/(Al,Rh^{3+})$ = 4 as atomic ratio, 2 M) in deionized water. Coprecipitation occurred by the dropwise addition of this solution into a $Na_2CO_3$ aqueous solution (1 M) at a constant temperature of 60 °C and pH 10. During the synthesis, the pH value was kept constant by addition of a NaOH aqueous solution (3 M). After the aging of the solid for 60 min, the obtained precipitate was filtered and washed with deionized water to obtain a residual $Na_2O$ content below the 0.01 wt %. The solid was then dried at 70 °C overnight and calcined at 900 °C for 6 h. When incipient wetness impregnation (IWI) was used to incorporate nickel and rhodium on the catalyst support, the metal nitrates aqueous solution was added to the coprecipitated HT support calcined at 900 °C, composed by MgO and $MgAl_2O_4$ and with a $Mg^{2+}/Al^{3+}$ atomic ratio of 4 to attain a final load of Ni and Rh of 3 and 0.5 wt %, respectively. The obtained solid was dried at 70 °C, (overnight) and then calcined at 900 °C for 6 h [52,63]. The synthesis of the Ni/Rh cluster used to disperse the active metals on the surface of the catalytic support is widely described elsewhere [64]. In this case, a proper amount of the $\{NEt_4\}_3\{Ni_{10}Rh(CO)_{19}\}$ cluster (Figure S1) to reach a final load of 2.0, 3.0 or 5.0 wt % of Ni (and consequently 0.3, 0.5 or 0.9 wt % of Rh) was maintained in inert atmosphere and dissolved in $CH_3CN$ (20 mL). When the catalytic support as powder was added to the solution, the obtained suspension was stirred for 12 h and then dried. The decomposition of the deposited cluster was carried out by a thermal treatment of the solid at 160 °C for 8 h. To confirm the deposition/decomposition of the cluster, infrared (IR) analyses were carried out (Figure S2). The obtained solid was then calcined at 900 °C for 6 h. The name and the composition of the catalysts are reported in Table 3.

**Table 3.** Catalyst nomenclature, precursor, and respective metal loading. C: cluster; WI: wet impregnation; IWI: incipient wetness impregnation; CP: coprecipitation of a hydrotalcite-type precursor.

| Catalyst | Precursor/Synthesis | Ni (wt %) | Rh (wt %) |
|---|---|---|---|
| 2NiRh C | $Ni_{10}Rh(CO)_{19}$/WI | 2.0 | 0.3 |
| 3NiRh C | $Ni_{10}Rh(CO)_{19}$/WI | 3.0 | 0.5 |
| 5NiRh C | $Ni_{10}Rh(CO)_{19}$/WI | 5.0 | 0.9 |
| 3NiRh IWI | $Ni(NO_3)_2$, $Rh(NO_3)_3$/IWI | 3.0 | 0.5 |
| 3NiRh CP | $Ni(NO_3)_2$, $Rh(NO_3)_3$/CP | 3.0 | 0.5 |
| Mg/Al/O (support) | $Mg(NO_3)_2$, $Al(NO_3)_3$/CP | / | / |

*3.2. Characterization Techniques*

The crystalline structure of the solids was determined by XRD powder analyses. The instrument used was a PANalytical X'Pert diffractometer (Spectris, Egham, UK) equipped with a copper anode ($\lambda C_{u\,K\alpha}$ = 0.15418 nm) and a fast X'Celerator detector (2θ range: 5–80°, step size: 0.1°, time per step: 2 s).

The specific surface area values of the catalysts were determined by BET method. The instrument used was a Carlo Erba Sorpty instrument (CARLO ERBA, Milano, Italy) [65]. The catalysts (~0.2 g) were pre-treated under vacuum at 150 °C for 30 min and analyzed at −196 °C (liquid $N_2$).

The $H_2$-TPR and $CO_2$-TPD experiments were carried out in a Autochem II (Chemisorption analyzer, Micromeritics) (Micromeritics, Norcross, GA, USA). In a typical $H_2$-TPR analysis, the calcined catalyst (~0.1 g) was pre-treated at 550 °C for 1 h under 30 mL min$^{-1}$ of He. After cooling of the sample to 50 °C, the carrier gas was switched to 5% (*v/v*) $H_2$/Ar (30 mL min$^{-1}$) and the temperature of the catalyst increased to 900 °C (10 °C min$^{-1}$), holding this temperature for 1 h. The signal was measured by means of a thermal conductivity detector (TCD). The $CO_2$-TPD experiments were carried out after the pretreatment of the solid at 500 °C (He, 30 mL min$^{-1}$). The $CO_2$ adsorption was carried out by flowing a constant flow of 10% (*v/v*) $CO_2$/Ar (30 mL min$^{-1}$) mixture for 2 h. The physiosorbed $CO_2$ was removed by using a flow of He (1 h). The desorbed $CO_2$ was monitored by heating up the solid to 500 °C (10 °C min$^{-1}$) under a stream of He. The $CO_2$ evolved was recorded through a Cirrus 2 (MKS Instruments) online mass spectrometer (MKS Instruments, Andover, MA, USA).

A TEM/STEM FEI TECNAI F20 microscope (Thermo Fisher Scientific, Waltham, MA, USA) equipped with an EDX analyzer was utilized for the high-resolution transmission electron microscopy (HRTEM) characterization. The solid was deposited on an Au grid with lacey multi-foil carbon film and the statistical distribution of the formed metal nanoparticles was determined by measuring the diameter of at least 200 particles [66].

The instrument used for the Raman spectroscopy analyses of the catalysts after reaction was a Renishaw RM1000 instrument (Renishaw plc, Wotton-under-Edge, UK). The beam laser used was green (Ar$^+$ 514.5 nm, relative power used: 10%) and the analyses were conducted in the 2000–800 cm$^{-1}$ region (acquisition time: 10 s, 10 accumulations) [67,68].

*3.3. Catalytic Activity*

The catalysts after calcination (powder) were pressed at 10 t for 20 min and subsequently crushed and sieved to a dimensional range of 30–40 mesh (0.420–0.595 mm). The catalysts (~1 g) were charged in a tubular reactor (INCOLOY 800 HT, ∅ = 0.8 cm) placed into an electric tubular furnace. The process gasses ($CH_4$, $CO_2$, $N_2$, $H_2$) flow rate was adjusted by thermal mass flowmeters (BROOKS Instruments, Hatfield, PA, USA) while water was fed to the reactor using an HPLC pump (Jasco, Tokyo, Japan) and a steam generator (T: 215 °C). The outlet stream was sent to a gas liquid separator operating at 1 °C followed by a back-pressure regulator (Swagelok, Solon, OH, USA). The analysis section consisted of a gas flowmeter (Figure 9) and an online gas chromatograph (Agilent Technologies 7890A, Santa Clara, CA, USA) equipped with a CarboPLOT P7 (carried gas $H_2$, determination of CO, $CO_2$ and $CH_4$) and HP-Molesieve (carrier gas $N_2$, determination of $H_2$) columns and two TCD.

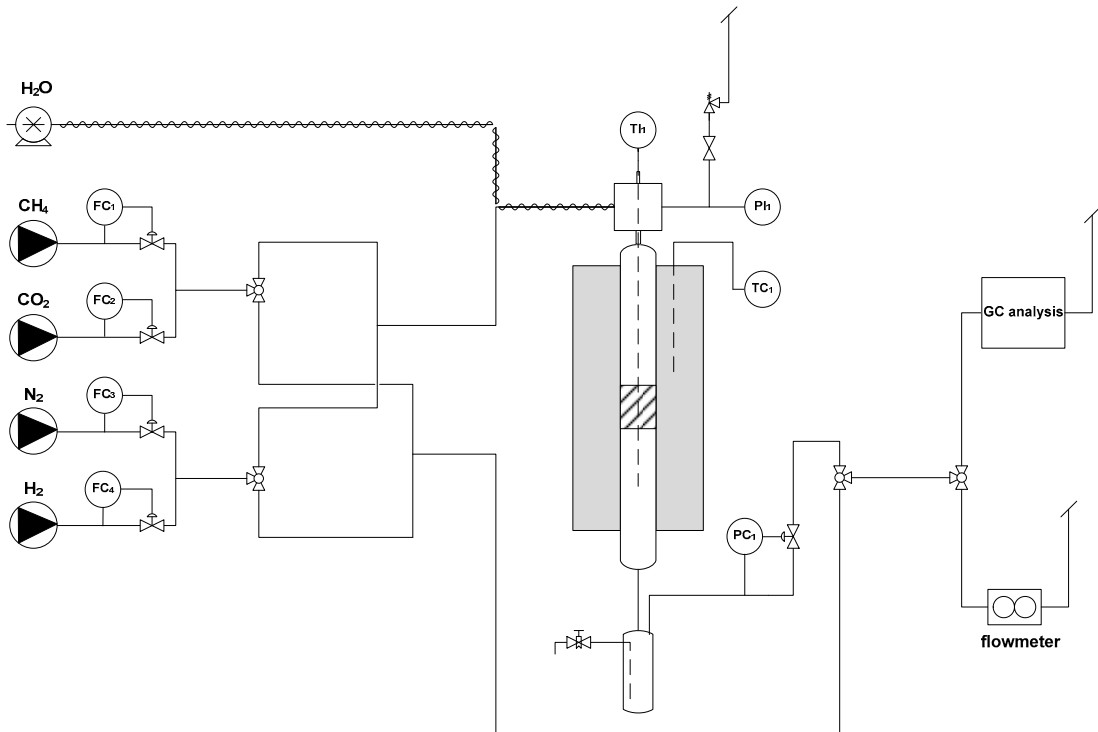

**Figure 9.** Scheme of the experimental setup used for the evaluation of the catalysts. FC: flow controller; TI: temperature indicator; TC: temperature controller; PI: pressure indicator; PC: pressure controller.

The activation of the catalysts was performed in-situ before the catalytic tests by continuously flowing of $H_2/N_2$ (1/10 *v/v*, 200 mL min$^{-1}$) increasing the catalyst temperature from 300 to 900 °C with a slope of 5 °C min$^{-1}$, and holding the final temperature for 1 h. After reduction, the catalysts were tested in the combined reforming reaction at 0.5 MPa, feeding a 1/1 (*v/v*) mixture of $CH_4$ and $CO_2$ and working at a constant GHSV of 50,000 mL h$^{-1}$ g$_{cat}$$^{-1}$. The catalytic activities were measured by changing the temperature of the oven from 900 to 700 °C and using steam to methane ratios (S/$CH_4$) values of the inlet stream of 2.0, 1.0 and 0.5 (*v/v*) (Table 4). Using a thermocouple sliding a capillary inserted into the catalytic bed it was possible to monitor the thermal profile during reaction.

**Table 4.** Operative conditions used for the evaluation of the catalysts.

| Temperature (°C) | S/$CH_4$ (*v/v*) | P (MPa) | GHSV (mL h$^{-1}$ g$_{cat}$$^{-1}$) |
|:---:|:---:|:---:|:---:|
| | 2.0 | | |
| 900 | 1.0 | | |
| | 0.5 | | |
| | 2.0 | 0.5 | 50,000 |
| 800 | 1.0 | | |
| | 0.5 | | |
| 700 | 2.0 | | |

The $CH_4$ and $CO_2$ conversion values were calculated using the following equation:

$$\text{Conversion} = \frac{(F_{in} \times y_{in}) - (F_{out} \times y_{out})}{(F_{in} \times y_{in})} \times 100$$

where F was the $CH_4$ or $CO_2$ flow rate and y was the molar fraction of $CH_4$ or $CO_2$ in the inlet ($_{in}$) or outlet ($_{out}$) stream. The calculations of the conversion values at the thermodynamic equilibrium were carried out using the CEAgui software by NASA [69].

## 4. Conclusions

The combined reforming of clean biogas using Ni–Rh/Mg/Al/O catalysts is a promising way to valorize $CO_2$, producing a syngas directly suitable for further downstream applications such as methanol or Fischer–Tropsch synthesis. Through the use of a pre-formed bimetallic cluster precursor, it was possible to obtain an optimized composition with low metal loading (down to 3.0 wt % of Ni) and to emphasize the interaction between Ni and Rh, promoting the formation of highly active and stable sites.

After reduction and reaction, the catalysts showed a high dispersion of the active phase, composed by a Ni–Rh alloy, highly active and stable in the harsh reaction conditions. When the metal loading was increased beyond optimum value, a dramatic decrease of activity was observed ascribable to the sintering of the active phase, with the formation of large agglomerates, mainly composed by Ni.

**Supplementary Materials:** The following are available online at http://www.mdpi.com/2073-4344/10/11/1345/s1, Figure S1: Structure of the {$Ni_{10}Rh(CO)_{19}$} cluster, Figure S2: IR spectra of {$Ni_{10}Rh(CO)_{19}$} before and after thermal decomposition, Figure S3: $H_2$/CO ratio of the outlet stream as a function of the reaction conditions, Figure S4: XRD patterns of the catalysts after reaction, Figure S5: Raman results obtained for the spent catalysts.

**Author Contributions:** Conceptualization, N.S., C.L., A.V. and M.C.I.; validation, N.S., C.L. and A.V.; formal analysis, A.C.; investigation, N.S., A.C., M.C.I.; writing—original draft preparation, N.S. and C.L.; writing—review and editing, A.V., N.S., C.L. and G.F.; supervision, G.F. and A.V.; All authors have read and agreed to the published version of the manuscript.

**Funding:** This research received no external funding.

**Conflicts of Interest:** The authors declare no conflict of interest.

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
