# Peer review of "Combined Reforming of Clean Biogas over Nanosized Ni–Rh Bimetallic Clusters"

_catalysts, doi:10.3390/catal10111345_

Round 1
Reviewer 1 Report
In this work, combined Steam/Dry Reforming of clean biogas was performed on Ni/Rh based catalysts. A variety of reaction conditions was covered and dispersed nanoparticles were produced. This article is quite interesting; it has a sufficient load of experimental work and provides new information. Moreover, it adheres to the journal's standards as it is an original paper on catalysis with an emphasis on the understanding of new catalytic materials with potential practical applications. Therefore my recommendation is to be accepted after applying some minor revisions.
- The effect of water in the dry/steam combined reforming was also witnessed at lower temperature by
le Saché, L. Pastor-Pérez, V. Garcilaso, D.J. Watson, M.A. Centeno, J.A. Odriozola, T.R. Reina, Flexible syngas production using a La2Zr2-xNixO7-δ pyrochlore-double perovskite catalyst: Towards a direct route for gas phase CO2 recycling, Catalysis Today, 357, 2020, 583-589, https://doi.org/10.1016/j.cattod.2019.05.039
It appears that CO2 conversion in the presence of steam is thermodynamically limited (due to competitive reactions). Could the author comment on the performance of their catalysts relative the thermodynamic equilibrium values?
- Authors claim in line 364 that the Ni10Rh(CO)19 cluster assured a better interaction between Ni and Rh, and helped preventing sintering. However, in view of the data presented just above, both Ni/Rh precursors led to similar particle sizes and large clusters. Please explain clearly how the data show this.
Overall I did not find clear the effect of this new cluster on the stability and performance of the catalyst compared to the IWI and CP samples. Maybe a summary would be helpful.
Author Response
In this work, combined Steam/Dry Reforming of clean biogas was performed on Ni/Rh based catalysts. A variety of reaction conditions was covered and dispersed nanoparticles were produced. This article is quite interesting; it has a sufficient load of experimental work and provides new information. Moreover, it adheres to the journal's standards as it is an original paper on catalysis with an emphasis on the understanding of new catalytic materials with potential practical applications. Therefore my recommendation is to be accepted after applying some minor revisions.
The authors are grateful for the reviewer positive feedback and happy to clarify the question raised.
1. The effect of water in the dry/steam combined reforming was also witnessed at lower temperature by
le Saché, L. Pastor-Pérez, V. Garcilaso, D.J. Watson, M.A. Centeno, J.A. Odriozola, T.R. Reina, Flexible syngas production using a La2Zr2-xNixO7-δ pyrochlore-double perovskite catalyst: Towards a direct route for gas phase CO2 recycling, Catalysis Today, 357, 2020, 583-589, https://doi.org/10.1016/j.cattod.2019.05.039
Responses: The authors thank the reviewer for providing such an interesting reference. We have added a comparative table in the main text in which our results are compared with the one suggested by the reviewer as well with other works from literature.
It appears that CO2 conversion in the presence of steam is thermodynamically limited (due to competitive reactions). Could the author comment on the performance of their catalysts relative the thermodynamic equilibrium values?
Responses: Yes, CO2 conversion is limited when water is added to the inlet feed. We have added the thermodynamic equilibrium values for both CO2 and CH4 conversion as a function of the conditions used in the main text. The calculated values are reported in figure 5 a,b, and further commented in the paragraph.
2. Authors claim in line 364 that the Ni10Rh(CO)19 cluster assured a better interaction between Ni and Rh, and helped preventing sintering. However, in view of the data presented just above, both Ni/Rh precursors led to similar particle sizes and large clusters. Please explain clearly how the data show this.
Overall I did not find clear the effect of this new cluster on the stability and performance of the catalyst compared to the IWI and CP samples. Maybe a summary would be helpful.
Responses: As pointed out by the reviewer, it is true that the catalysts exhibited a portion of the formed nanoparticles with a larger diameter and a quite similar particle size distribution but, it is also important to clarify that:
- The activity of the particles is a function of their size, smaller the diameter, higher the activity in the reaction. Consequently, the higher number of nanoparticles with a diameter centred around 2nm (c.a. 50% of the total) found for the 3NiRh C catalyst, play a crucial role in the catalytic activity and suggests a higher interaction between Ni and Rh (that improve the dispersion of the active phase).
- Differently from IWI deposition (random distribution of nanoparticles over the catalyst surface), it was found that using the Ni-Rh cluster as a precursor, the disposition of nanoparticles over the catalyst support occurred preferentially in kink positions, that are well-known to be very active sites.
The differences observed in the catalytic activity (especially at 800°C) can be considered marked. It is noteworthy that even few improvements in percentage conversion (even 5-6%) can highly impact, from the point of view of an industrial plant, in both terms of productivity and set-up costs.
Reviewer 2 Report
Journal: Catalysts
Manuscript ID: catalysts-1002071
Type of manuscript: Article
Title: Combined Reforming of Clean Biogas over Nanosized Ni-Rh Bimetallic Clusters
The catalysts preparation is well explained as well as the characterization techniques. The English grammar is good and does not need improvement. The results are sound and well explained. Comparison to literature data is missing. Please consider the following points, after which publication is advisable.
- Page 3: The “range 30-40 mesh”, is this a size measure of the mesh (30-40 μm)?
- From Table 2 and from the text it seem that only 1 GHSV was used. Is enough for a catalytic process study?
- Please use dot instead of comma for the decimal separator (Figure 1).
- On Figure 4c the length label is half-visible.
- A scheme of the experimental set-up and the chemical equations under considerations would be informative for the reader.
- The discussion should be made also with comparison to the cited literature data. The authors claim the activity is remarkable, but no comparison is provided.
- A comparison of the activity of the authors’ catalyst and other published catalysts would be appropriate.
- On page 2 there is a mistake “0catalytic”.
- In the Supplementary Figure S2 is very poor.
Author Response
The catalysts preparation is well explained as well as the characterization techniques. The English grammar is good and does not need improvement. The results are sound and well explained. Comparison to literature data is missing. Please consider the following points, after which publication is advisable.
1. Page 3: The “range 30-40 mesh”, is this a size measure of the mesh (30-40 μm)?
We provided to add the value in mm in the text (line 338). 30-40 mesh = 0.420-0.595 mm
2. From Table 2 and from the text it seem that only 1 GHSV was used. Is enough for a catalytic process study?
We agree with the reviewer that the GHSV affects the catalytic performances. In the manuscript the different catalysts have been compared in order to find not only the best catalyst but to understand and confirm the role of chemical promoter on the active phase. To do that, it is important to verify the rate of conversion and how the different materials evolve under different thermal and hydrothermal conditions. We expect that the conversion rate moves in the opposite direction of the GHSV. Considering that for a process interesting from an industrial point of view, it is important not only the conversion but also the amount of biogas processed per unit of time and weight of catalysts, we decided to compare all the catalyst with a GHSV that could maximize both the parameters and in the range of that used in the industrial steam reforming on natural gas. Focusing the attention of the results reported in this manuscript, working at high temperature (industrial) the catalysts reach the equilibrium value then a decrease of GHSV does not affect the conversion. Working at lower temperature and S/CH4 ratio may be possible that decrease of GHSV could enhance the conversion.
3. Please use dot instead of comma for the decbimal separator (Figure 1).
Figure 1 was modified as requested.
4. On Figure 4c the length label is half-visible.
The label in figure 4 was opportunely modified.
5. A scheme of the experimental set-up and the chemical equations under considerations would be informative for the reader.
We agree with the reviewer comment. We added and explained a scheme of the set-up in the manuscript (section 3: materials and methods line 347). We also added chemical equations and equation for the conversion calculus.
6. The discussion should be made also with comparison to the cited literature data. The authors claim the activity is remarkable, but no comparison is provided.
7. A comparison of the activity of the authors’ catalyst and other published catalysts would be appropriate.
We fully agree with the reviewer comments. We added a section (from line 212) in which catalytic results obtained in this work have been compared and discussed with literature data. In order to compare the data, results were normalized in terms of amount of biogas converted per unit of time and mass of active phase.
8. On page 2 there is a mistake “0catalytic”.
The mistake has been corrected.
9. In the Supplementary Figure S2 is very poor.
The quality of figure S2 was increased, as requested.
Reviewer 3 Report
This work is well-done, well-organised and the results have the potential to be interesting and useful. The experiments are well designed and the results are meaningful. I recommend publishing this manuscript. From my point of view, there are some questions that should be addressed before its publication.
- From the Table 2. did you try to use other GHSV? Is there any information what happens before or after 50,000 mL h-1 gcat-1 GHSV?
- there is ar really high valued BET surface reduction followed by the catalytic reactions. What is the thermal stability of these catalysts? For how long can be they used without any degradation?
- An XPS is needed to understand what is the oxidation states of the metals ont he surface, or other technique to understand whaat is the active sites during this reaction.
Author Response
This work is well-done, well-organised and the results have the potential to be interesting and useful. The experiments are well designed and the results are meaningful. I recommend publishing this manuscript. From my point of view, there are some questions that should be addressed before its publication.
- From the Table 2. did you try to use other GHSV? Is there any information what happens before or after 50,000 mL h-1 gcat-1 GHSV?
To compare the different catalysts in this manuscript we used a GHSV of 50 L/h*gcat because it is a good compromise between the rate of conversion achievable and the amount of biogas processable per unit of time and unit of mass of catalyst. Moreover, this value is in the range of the GHSV used for the industrial steam reforming of natural gas. Analysing the data reported in this work it could be possible to observe that working at high temperature (similar to the industrial process) the conversion approaches the equilibrium, then a reduction of GHSV decreases the productivity. Working at lower temperature and lower S/CH4 ratio a decrease of GHSV may cause an increase of conversion.
- there is ar really high valued BET surface reduction followed by the catalytic reactions. What is the thermal stability of these catalysts? For how long can be they used without any degradation?
As pointed out by the reviewer it is true that the catalysts after reaction showed a decrease in their specific surface area. It is noteworthy that, part of this decrease is attributable to the Ni and Rh-oxides reduction after the activation step, that causes the loss of part of the catalyst porosity. Other losses could be due to the harsh hydrothermal conditions used in the catalytic tests. In this work the catalyst activities were investigated over a range of conditions that, in fact, simulate a rapid aging of the samples. Considering that the catalysts showed no deactivation in terms of biogas conversion, it could be asserted that the loss of specific area observed, may not be a factor that affects the catalysts stability. Moreover, the aim of this work was not to demonstrate a long-term stability of the catalysts (that requires activity tests of at least 250-500 h continuously) but to compare the effect of different Ni and Rh incorporation methods. Anyway, considering that the catalytic support used was not so different to the one applied in the industrial process, we could aspect that these catalysts may be stable even for longer time on stream.
- An XPS is needed to understand what is the oxidation states of the metals ont he surface, or other technique to understand whaat is the active sites during this reaction.
It is true that through XPS analyses it would be possible to determinate the oxidation state of the active surface. It is also true that as demonstrated in our previous works (reference 52 in the manuscript and Benito, Dal Santo et al. applied catalysis B 179 (2015) 150-159) a Ni/Mg/Al catalyst in steam reforming condition (high value of S/CH4 ratio) easily undergoes to re-oxidation strongly affecting the catalysts activity/stability. The addition of a promoter (such as Rh) stabilizes the Ni0 particles, avoiding re-oxidation and thus improving the catalyst activity (as pointed out also by H2-TPR (Kugai et al. Journal of catalysis 238 (2006) 430-404) and HRTEM analyses in the work). Starting from these results it is possible to claim that the active phase for the reaction is composed by Ni0, Rh0 and Ni-Rh sites. To further demonstrate this and the nature of the metal-support interactions, an in-operando XAS analysis would be required.